# Effects of Biologic Therapy on Laboratory Indicators of Cardiometabolic Diseases in Patients with Psoriasis

**DOI:** 10.3390/jcm12051934

**Published:** 2023-03-01

**Authors:** Teppei Hagino, Hidehisa Saeki, Eita Fujimoto, Naoko Kanda

**Affiliations:** 1Department of Dermatology, Nippon Medical School Chiba Hokusoh Hospital, Inzai 270-1694, Japan; 2Department of Dermatology, Nippon Medical School, Tokyo 113-8602, Japan; 3Fujimoto Dermatology Clinic, Funabashi 274-0063, Japan

**Keywords:** biologics, cardiometabolic disease, tumor necrosis factor-α inhibitor, high-density lipoprotein-cholesterol, psoriasis, uric acid

## Abstract

Psoriasis is associated with cardiometabolic and cardiovascular diseases. Biologic therapy targeting tumor necrosis factor (TNF)-α, interleukin (IL)-23, and IL-17 may improve not only psoriasis but also cardiometabolic diseases. We retrospectively evaluated whether biologic therapy improved various indicators of cardiometabolic disease. Between January 2010 and September 2022, 165 patients with psoriasis were treated with biologics targeting TNF-α, IL-17, or IL-23. The patients’ body mass index; serum levels of HbA1c, total cholesterol, high-density lipoprotein-cholesterol (HDL-C), low-density lipoprotein-cholesterol, triglyceride (TG), and uric acid (UA); and systolic and diastolic blood pressures were recorded at weeks 0, 12, and 52 of the treatment. Baseline psoriasis area and severity index (week 0) positively correlated with TG and UA levels but negatively correlated with HDL-C levels, which increased at week 12 of IFX treatment compared to those at week 0. UA levels decreased at week 12 after ADA treatment compared with week 0. HDL-C levels decreased 52 weeks after IXE treatment. In patients treated with TNF-α inhibitors, HDL-C levels increased at week 12, and UA levels decreased at week 52, compared to week 0. Thus, the results at two different time points (at weeks 12 and 52) were inconsistent. However, the results still indicated that TNF-α inhibitors may improve hyperuricemia and dyslipidemia.

## 1. Introduction

Psoriasis is a chronic inflammatory skin disease that presents with scaly indurated erythema. The tumor necrosis factor (TNF)-α/interleukin (IL)-23/IL-17 axis is the mainstay of psoriasis pathogenesis. Biologics targeting TNF-α, IL-23, and IL-17 have been developed and have shown significant therapeutic effects in psoriasis. Psoriasis is frequently associated with cardiometabolic diseases such as diabetes mellitus (DM), dyslipidemia (DL), hyperuricemia (HUA), hypertension (HT), and ischemic heart disease, suggesting a pathogenetic link between psoriasis and cardiometabolic diseases [1].

Approximately 40% of patients with psoriasis are obese and develop cardiovascular diseases (CVDs) more frequently than healthy individuals [2,3]. Cytokines, TNF-α, IL-23 and IL-17 are known to be involved in the development of cardiometabolic diseases [4]. Eleven biologics have been approved for psoriasis in Japan, targeting TNF-α (adalimumab [ADA], infliximab [IFX], certolizumab pegol [CZP]), IL-17 (ixekizumab [IXE], secukinumab [SEC], brodalumab [BRO], bimekizumab), and IL-23 (ustekinumab [UST], guselkumab [GUS], risankizumab [RIS], and tildrakizumab). These biologics may improve not only the rash of psoriasis, but also comorbid cardiometabolic diseases. In this study, we retrospectively evaluated whether treatment with these biologics improved the values of laboratory or clinical indicators of cardiometabolic diseases.

## 2. Materials and Methods

### 2.1. Study Design and Data Collection

This study was conducted in accordance with the Declaration of Helsinki (2004) and approved by the Ethics Committee of Nippon Medical School Chiba Hokusoh Hospital. From January 2010 to September 2022, 165 Japanese patients with psoriasis (aged ≥ 18 years; 131 males and 34 females) were treated with any of the biologics (ADA, IFX, CZP, IXE, SEC, BRO, UST, GUS, and RIS) for more than 52 weeks at the outpatient clinic.

In this study, we excluded the patients who had been treated with any of the biologics but stopped the treatments earlier than 52 weeks. The patients included both biologic naïve patients and those who switched from other biologics (regardless of how long they had been treated previously).

Disease severity and laboratory or clinical indicators of cardiometabolic diseases of the patients were recorded during therapy and retrospectively analyzed. Written informed consent was obtained from all the patients. The diagnosis of psoriasis was made by dermatologists based on clinical symptoms and progress. Patient age, body mass index (BMI), disease duration, presence or absence of arthritis, DM, HT, DL, HUA, CVD, and current smoking status were examined before treatment.

Psoriasis area and severity index (PASI), BMI, serum HbA1c, total cholesterol (TC), high-density lipoprotein-cholesterol (HDL-C), low-density lipoprotein-cholesterol (LDL-C), triglyceride (TG), uric acid (UA), and systolic/diastolic blood pressure (sBP/dBP) levels were analyzed at weeks 0, 12, and 52 of treatment. The percentage of patients with a PASI decrease of 75% or more, 90% or more, or 100% or more from the baseline (PASI 75, PASI 90, or PASI 100, respectively) was calculated at weeks 12 and 52.

### 2.2. Statistical Analysis

All statistical analyses were performed using EZR software (Jichi Medical School, Saitama Medical Center). The normality of the data distribution was assessed using the Shapiro–Wilk test. Variables with a normal distribution are expressed as a mean ± standard deviation, and those with a nonparametric distribution are expressed as median [interquartile range]. Correlation analysis was performed using Spearman’s correlation coefficient.

Differences between weeks 0, 12, and 52 were analyzed by repeated measures analysis of variance for variables with a normal distribution and Friedman’s test for variables with a nonparametric distribution. Post hoc analyses were performed using Bonferroni correction. Statistical significance was set at *p* < 0.05.

## 3. Results

### 3.1. Demographics of Patients with Psoriasis

One hundred and sixty-five Japanese patients with psoriasis (131 men and 34 women) were enrolled in this study (Table 1). The TNF-α inhibitors IFX, ADA and CZP were administered to 28, 17, and 11 patients, IL-17 inhibitors; IXE, SEC, and BRO to 38, 13 and 12 patients; and IL-23 inhibitors, UST, GUS, and RIS to 10, 19, and 17 patients, respectively.

In terms of the overall male/female ratio, the result was a higher proportion of males (approximately 80%). Overall, 80 patients were diagnosed with arthritis. Patients with arthritis were more likely to choose TNF-α inhibitors. At baseline, 20, 43, 25, 25, and 6 patients had diabetes, hypertension, dyslipidemia, hyperuricemia, and cardiovascular disease, respectively. A total of 98 patients were smokers. In terms of individual biologics, IXE had the highest number of cases (Table 1). All the patients in the CZP-treated group had arthritis. Risankizumab was selected for patients with pre-existing cardiovascular diseases. Patients administered ADA had the highest smoking rate.

### 3.2. Correlations of PASI with Indicators of Cardiometabolic Diseases before Treatment

Before treatment, the correlations between PASI and individual indicators of cardiometabolic diseases were analyzed (Table 2). The baseline (week 0) PASI was negatively correlated with HDL-C levels and positively correlated with TG and UA levels (Table 2). The other indicators showed no significant correlations with PASI.

### 3.3. Changes in Indicators of Cardiometabolic Diseases after Treatment with TNF-α Inhibitors

After IFX treatment, the HDL-C value increased significantly at week 12, while the value at week 52 was not altered compared with baseline (Table 3). After ADA treatment, the UA value decreased significantly at week 12, whereas the value at week 52 was not altered compared to baseline (week 0). The values of laboratory and clinical indicators examined did not change after CZP treatment compared with the baseline values (week 0). After treatment with all TNF-α inhibitors, the HDL-C value increased significantly at week 12, while the value at week 52 was not altered compared with baseline (week 0). After treatment with all TNF-α inhibitors, the UA value decreased significantly at week 52, whereas the value at week 12 was not altered compared with baseline (week 0). The values of the other indicators were not altered after treatment with TNF-α inhibitors.

### 3.4. Changes in Indicators of Cardiometabolic Diseases after Treatment with IL-17 Inhibitors

After IXE treatment, the HDL-C value decreased significantly at week 52 compared to baseline (week 0) (Table 4). After BRO and SEC treatment, the values of laboratory and clinical indicators examined did not change compared to the baseline values. After treatment with all IL-17 inhibitors, the values of laboratory and clinical indicators examined did not change compared to baseline.

### 3.5. Changes in Indicators of Cardiometabolic Diseases after Treatment with IL-23 Inhibitors

After treatment with GUS, UST, RIS, and all IL-23 inhibitors, the values of laboratory and clinical indicators examined did not change compared with baseline (week 0) (Table 5).

### 3.6. The Improvement of PASI by TNF-α, IL-17, or IL-23 Inhibitors

The PASI 75, PASI 90, and PASI 100 scores after treatment with individual biologics are shown in Figure 1. In PASI 75 and PASI 90, IL-17 inhibitors (SEC, IXE, BRO) and IL-23p19 antibodies (GUS, RIS) were the most effective, while TNF-α inhibitors (ADA, IFX, CZP) and IL-12/23p40 antibody (UST) showed lower efficacy. In PASI 100, BRO was the most effective, followed by IXE, with slightly lower efficacy in SEC, IL-23p19 antibodies (GUS, RIS) and p40 antibody (UST), whereas TNF-α inhibitors (ADA, IFX, CZP) showed much lower efficacy. Overall, IL-17 inhibitors (SEC, IXE, BRO) and IL-23p19 antibodies (GUS, RIS) were the most effective in improving PASI, while TNF-α inhibitors (ADA, IFX, CZP) and IL-12/23p40 antibody (UST) were inferior.

### 3.7. Correlation between Percent Reduction of PASI Versus Percent Changes of HDL-C or UA

We then analyzed whether the significant changes in HDL-C or UA by TNF-α or IL-17 inhibitors may correlate with the percent reduction in PASI by identical biologics. There was no significant correlation between the percent reduction of PASI and percent changes in HDL-C or UA by the identical biologics (Table 6).

## 4. Discussion

A total of 165 patients were included in the current analysis, which selected 11 biologic treatments used in Japan. There was a clear difference in sex, with 80% of the patients being male. In Japan, 65.8% of psoriasis patients are male, indicating a male predominance in the patient population [5]. This may also be characteristic of the limited area of Chiba.

Before treatment with biologics, PASI significantly correlated positively with serum TG and UA levels and negatively with serum HDL-C levels. Psoriasis skin lesions show hyperproliferation of keratinocytes, which increases the rate of DNA formation and purine synthesis and may lead to HUA [6]. Conversely, urate crystals stimulate keratinocytes to proliferate and produce inflammatory cytokines/chemokines, such as IL-1α or IL-8, indicating the promoting effects of UA on psoriasis [7]. Urate crystals activate the NLR family pyrin domain-containing three inflammasomes in macrophages, resulting in the production of active IL-1β and IL-18 [8], and promote the production of TNF-α in monocytes [9], indicating that UA might increase the levels of inflammatory cytokines promoting psoriasis. A previous study also reported a positive correlation between PASI and serum UA levels [10]. Proinflammatory cytokines, such as TNF-α, promote the production of very low-density lipoprotein-TG by activation of NF-κB in hepatocytes, thus increasing serum TG levels [11,12]. Tumor necrosis factor-α, IL-1, and IL-6 inhibit lipoprotein lipase activity in adipose tissues, thereby decreasing TG clearance and increasing the level of TG in plasma [13,14]. These reports indicate that pro-inflammatory cytokines that induce psoriasis may promote hypertriglyceridemia in parallel. In contrast, HDL is enriched in anti-inflammatory lipids, and interaction of HDL with immune cells induces cholesterol efflux from cell membranes, suppresses immune cell activation, acts on dendritic cells, suppresses their expression of CD40, 80, 86, or class II molecules, and reduces their ability to differentiate naïve T cells into Th1 or Th17 cells, indicating the protective effects of HDL from psoriasis [15]. Previous studies have also reported an association between psoriasis and high TG and low HDL-C levels [16]. The significant correlation between PASI and these cardiometabolic indicators indicates that psoriasis may not only be a skin-limited but also a systemic inflammatory disease.

HDL-C levels increased significantly at weeks 12 and 52 after treatment with IFX or TNF-α inhibitors, respectively. Previous reports have also shown that IFX increases HDL-C levels in patients with psoriasis [17]. TNF-α suppresses the expression of ATP-binding cassette transporter A1 (ABCA1), mediating the rate-controlling step of HDL formation in human intestinal Caco-2 cells, thus decreasing HDL-C levels [18]. Tumor necrosis factor-α inhibitors may counteract the inhibitory effects of TNF-α on HDL formation and thus increase HDL-C levels. HDL prevents the oxidation of LDL, induces efflux of accumulated LDL in the vasculature, and prevents foam cell accumulation, thereby suppressing atherosclerosis [19]. Decreased HDL-C levels are associated with the development of ischemic heart disease [20]. HDL also suppresses the production of inflammatory cytokines and chemokines, such as TNF-α, IL-1β, IL-6, IL-8, CCL3, and CCL4, in human monocytes induced by contact with stimulated T cells [21]. It has previously been reported that ADA treatment decreased serum levels of vascular cell adhesion molecule 1, a biomarker of cardiovascular diseases, in patients with plaque psoriasis [22], indicating the cardioprotective effects of ADA. However, regulatory T lymphocyte dysfunction is important in the development of atherosclerosis, and ADA treatment in patients with psoriasis vulgaris resulted in a decrease in plasma regulatory cytokines (IL-10, transforming growth factor-β1, and IL-35) [23], indicating the pro-atherosclerotic effects of ADA.

After treatment with ADA or TNF-α inhibitors, UA levels decreased significantly at weeks 12 and 52, respectively. Hepatic and plasma TNF-α induce c hepatic parenchymal cell injury, causing de novo purine synthesis and accelerating UA production [24]. Thus, TNF-α may systemically increase UA levels, which may be counteracted by TNF-α inhibitors. Uric acid promotes the progression of atherosclerosis by reducing NO production and inducing superoxide generation in endothelial cells [25]. Meta-analyses have shown a correlation between HUA and CVD risk [26,27]. Patients with psoriatic arthritis with HUA showed greater carotid intima-media thickness than normouricemic patients [28].

Although IFX, ADA, and CZP are all TNF-α inhibitors, the results among these biologics are inconsistent; CZP treatment did not alter HDL-C or UA levels, only ADA decreased UA levels, and only IFX increased HDL-C levels. The inconsistency may at least partially be due to the small sample size and uneven distribution; IFX, ADA, and CZP were administered to 28, 17, and 11 patients, respectively.

Temporarily increased HDL-C or decreased UA levels at week 12 returned to baseline at week 52. Therefore, it is unclear whether these changes will lead to protection against cardiometabolic diseases. In future, we should examine the levels of these indicators for a longer duration in a larger sample size.

HDL-C levels were reduced at week 52 after treatment with the anti-IL17A antibody IXE in patients with psoriasis. These results are contrary to our expectations and indicate that IL-17A may increase HDL levels. To date, conflicting results have been reported regarding the role of IL-17A in cardiovascular and/or metabolic diseases; IL-17A has both proatherogenic and atheroprotective effects [29]. Th17-polarized cells from non-obese diabetic mice following mycobacterial adjuvant immunotherapy delay the development of type 1 diabetes [30], indicating the protective effects of IL-17A against diabetes. It has been reported that metabolic syndromes [31], severe coronary artery diseases [32], and atherosclerosis in rabbits [33] are associated with decreased serum IL-17A levels as well as decreased HDL-C levels, and the improvement of atherosclerosis in a rabbit model leads to an increase in both levels [33], indicating a positive correlation between HDL-C and IL-17A in cardiometabolic diseases. Interleukin-17A acts on mouse endothelial cells and enhances their expression of ABCA1 [34], a transporter mediating HDL synthesis, indicating the possible promoting effects of IL-17A on HDL synthesis. Further studies are required to elucidate the direct effects of IL-17A on HDL expression. Our results also indicate that physicians should use IL-17A inhibitors with caution in patients with psoriasis associated with cardiometabolic disease**.**

Several studies have indicated that IL-23 may increase serum LDL-C levels [35] or promote the development of DM [36]. It has been reported that anti-IL-12/23p40 antibody UST treatment reduced the size of the lipid-rich necrotic core, a high-risk coronary artery plaque in patients with psoriasis [37], while increasing the occurrence of major adverse cardiovascular events [38]. There are few studies on the effects of anti-IL-23p19 antibodies on metabolic or cardiovascular diseases, possibly because these are relatively new biologics. In the present study, IL-23 inhibitors as well as anti-IL-12/23p40 and IL-23p19 antibodies did not alter the levels of laboratory or clinical indicators of cardiometabolic diseases in patients with psoriasis. However, treatment with IL-23 inhibitors might alter the levels of different serum indicators or those on imaging, such as coronary CT angiography, and these should be further examined in a larger cohort of patients.

There was no significant correlation between percentage changes in HDL-C or UA and percentage reduction in PASI by TNF-α or IL-17 inhibitors. The results indicate that the effects of TNF-α or IL-17 inhibitors on UA or lipid metabolism may not always be parallel to those on skin rash in psoriasis. Although TNF-α inhibitors were less effective in the improvement of PASI than IL-17 and IL-23 inhibitors, they showed possible beneficial effects on cardiometabolic diseases. However, the improvement in HDL-C or UA by TNF-α inhibitors was transient, and the increase in HDL-C by IFX or decrease in UA by ADA at week 12 was not maintained at week 52. It is possible that the effects of these biologics on lipid or UA metabolism may be unstable and likely to be disturbed by other factors, such as diet, medicines, and other comorbid diseases such as infection. We should further examine the effects of biologics on cardiometabolic indicators over a longer duration (up to five years) and in a larger cohort.

In the present study, we did not investigate new occurrences of DM, HTN, DL, or HUA during the study period. Further studies should investigate the occurrence of these diseases in a larger cohort over a longer duration.

This study had several limitations. First, it was a retrospective study involving a small number of patients. Second, the number of patients for each biologic was biased because of the preferable usage of more effective biologics. Additionally, this study included both bio-naïve and bio-switched patients. We should further examine the influence of prior biologic treatment on the levels of indicators of cardiometabolic diseases in a larger cohort. To compare the effects of different biologics on cardiometabolic indicators, patients should be uniformly assigned to each biologic. Third, we cannot avoid the confounding effects of medications for DM, DL, HT, and HUA, such as insulin or 3-hydroxy-3-methylglutaryl-coenzyme A reductase inhibitors, since some of the patients were taking these medicines during treatment with biologics. Further studies should be conducted to exclude patients taking such medication.

## 5. Conclusions

In conclusion, baseline PASI was significantly positively correlated with serum TG and UA levels and negatively correlated with serum HDL-C levels. At week 12 of IFX treatment, HDL-C level increased, and at week 12 of ADA treatment, UA level decreased compared to week 0. At week 52 of IXE treatment, HDL-C levels decreased compared to those at week 0. After treatment with all TNF-α inhibitors, HDL-C levels increased at week 12, and UA levels were reduced at week 52 compared to week 0. These results indicate that TNF-α inhibitors may improve HUA and dyslipidemia in patients with psoriasis.

## Figures and Tables

**Figure 1 jcm-12-01934-f001:**
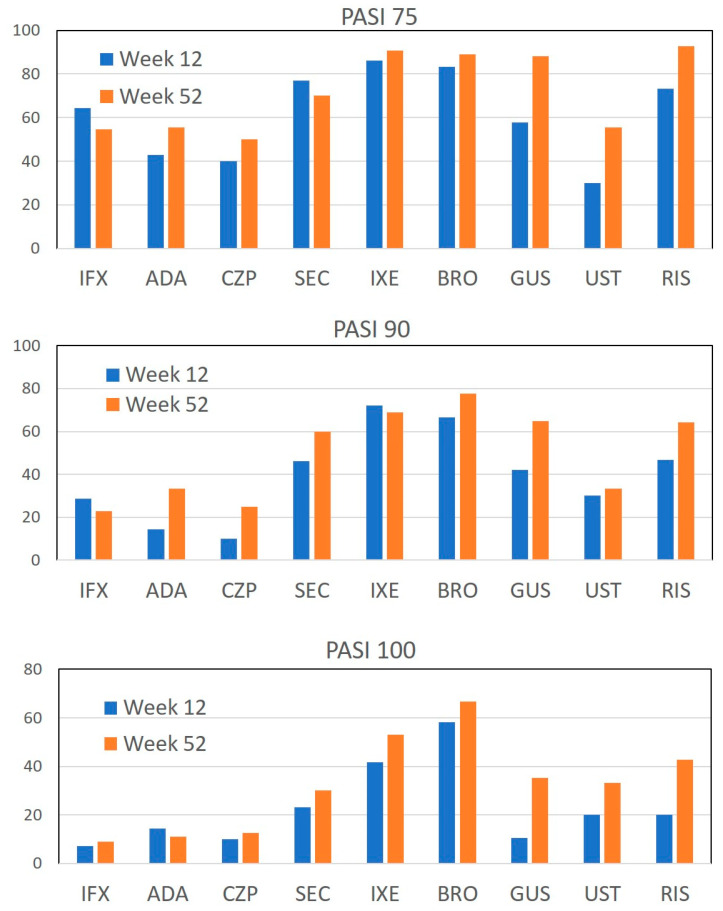
The achievement rates of psoriasis area and severity index (PASI) 75, 90, and 100 at week 12 and 52 of treatment with individual biologics in patients with psoriasis. IFX, infliximab; ADA, adalimumab; CZP, certolizumab pegol; SEC, secukinumab; IXE, ixekizumab; BRO, brodalumab; GUS, guselkumab; UST, ustekinumab; RIS, Risankizumab.

**Table 1 jcm-12-01934-t001:** (a) Baseline demographic and disease characteristics of patients with psoriasis in 3 groups of biologics; (b) Baseline demographic and disease characteristics of patients with psoriasis in each biologic.

**(a)**													
**Biologic (*n*)**			**All TNF-α Inhibitors (56)**			**All IL-17 Inhibitors (63)**			**All IL-23 Inhibitors (46)**			**All Biologics (165)**	
**Sex, *n* (%)**													
Male			44 (78.6)			50 (79.4)			37 (80.4)			131 (79.4)	
Age (years) ^b^			54 [43.75–68]			58 [45.5–68.5]			61 [47.5–72.75]			56 [45–69]	
Body mass index (kg/m^2^) ^a^			24.96 ± 4.11			25.45 ± 4.46			24.84 ± 3.65			25.15 ± 4.11	
Disease duration (years) ^b^			16.5 [9.25–27]			10 [5–20]			17.5 [8.25–26.5]			15 [6.5–25]	
Presence of arthritis, *n* (%)			37 (66.1)			30 (47.6)			13 (28.3)			80 (49.1)	
PASI ^b^			9.05 [5.1–12.6]			10.8 [8.3–13.55]			10.9 [7.95–14.35]			10.8 [7.2–13.6]	
Diabetes mellitus, *n* (%)			8 (14.3)			5 (7.9)			7 (15.2)			20 (12.3)	
Hypertension, *n* (%)			16 (28.6)			15 (23.8)			12 (26.1)			43 (26.1)	
Dyslipidemia, *n* (%)			7 (12.5)			11 (17.5)			7 (15.6)			25 (15.2)	
Hyperuricemia, *n* (%)			8 (14.3)			9 (16.1)			8 (17.4)			25 (15.2)	
Cardio vascular disease, *n* (%)			1 (1.85)			2 (3.23)			3 (6.52)			6 (3.7)	
Current smoking, *n* (%)			36 (72)			36 (60)			26 (61.9)			98 (64.5)	
**(b)**													
**Biologic (*n*)**	**IFX (28)**	**ADA (17)**		**CZP (11)**	**SEC (13)**		**IXE (38)**	**BRO (12)**		**UST (10)**	**GUS (19)**		**RIS (17)**
**Sex, *n* (%)**													
Male	22 (78.6)	14 (82.4)		8 (72.7)	11 (84.6)		29 (76.3)	12 (83.3)		8 (80)	17 (89.5)		12 (70.6)
Age (years) ^b^	55 [47–68.25]	54 [43–69]		53 [43.5–61]	63 [52–70]		56 [43.5–65.75]	63 [46–69.5]		50 [45.5–69]	66 [50–75.5]		61 [50–69]
Body mass index (kg/m^2^) ^a^	25.48 ± 4.36	23.8 ± 3.7		25.17 ± 4.05	25.37 ± 4.97		25.59 ± 4.72	25.13 ± 3.22		25.08 ± 3.48	25.33 ± 3.56		24.27 ± 3.84
Disease duration (years) ^a,b^	16.5 [10–27.75]	20 [11.5–26]		7 [4.5–28]	10 [5–30]		9 [4–15]	17.5 [10–26.25]		17.5 [12–23.75]	15 [7.5–31.5]		18 [6–25]
Presence of arthritis, *n* (%)	15 (53.6)	11 (64.7)		11 (100)	4 (30.8)		21 (55.3)	5 (41.7)		2 (20)	7 (36.8)		4(23.5)
PASI ^a,b^	11.7 [8.6–16.875]	7.8 [3.3–9.95]		6 [3.2–8.6]	10.8 [9–12.9]		11.3 [7.6–19.275]	9.4 [10.85–8.35]		8.5 [7.025–14.4]	10.8 [8.1–13.9]		11.4 [10.2–15.8]
Diabetes mellitus, *n* (%)	3 (10.7)	4 (23.5)		1 (9.1)	1 (7.7)		3 (7.9)	1 (8.3)		1 (10)	4 (21.1)		2 (11.8)
Hypertension, *n* (%)	8 (28.6)	6 (35.3)		2 (18.2)	4 (30.8)		7 (18.4)	4 (33.3)		1 (10)	4 (21.1)		7 (41.2)
Dyslipidemia, *n* (%)	5 (17.9)	2 (11.8)		0 (0)	1 (7.7)		7 (18.4)	3 (25)		0 (0)	4 (21.1)		3 (18.8)
Hyperuricemia, *n* (%)	3 (10.7)	2 (11.8)		3 (27.3)	3 (23.1)		5 (13.2)	1 (8.3)		1 (10)	5 (26.3)		2 (11.8)
Cardio vascular disease, *n* (%)	1 (3.6)	0 (0)		0 (0)	1 (8.3)		0 (0)	1 (8.3)		0 (0)	1 (5.3)		2 (11.8)
Current smoking, *n* (%)	17 (68)	12 (85.7)		7 (63.6)	9 (69.2)		22 (61.1)	5 (41.7)		5 (71.4)	11 (61.1)		10 (58.8)

(a): ^a^ Data provided as the mean ± standard deviation. ^b^ Data provided as the median [interquartile range]. PASI, psoriasis area, and severity index; (b): ^a^ Data provided as the mean ± standard deviation. ^b^ Data provided as the median [interquartile range]. PASI, psoriasis area and severity index; ADA, adalimumab; IFX, infliximab; CZP, certolizumab pegol; IXE, ixekizumab; SEC, secukinumab; BRO, brodalumab; UST, ustekinumab; GUS, guselkumab; RIS, risankizumab.

**Table 2 jcm-12-01934-t002:** Correlations of PASI with indicators of cardiometabolic diseases before treatment with biologics.

	Rho	*p*
Body mass index	0.0488	0.536
HbA1c	0.0945	0.239
Total cholesterol	0.083	0.308
HDL-C	−0.216	0.0163 *
LDL-C	0.0413	0.642
Triglyceride	0.201	0.0136 *
Uric acid	0.241	0.00207 **
Systolic blood pressure	−0.0406	0.627
Diastolic blood pressure	−0.0113	0.893

Correlations between variables were examined using Spearman’s correlation coefficients. * Statistically significant at *p* < 0.05, ** at *p* < 0.01. HDL-C, high-density lipoprotein-cholesterol; LDL-C, low-density lipoprotein-cholesterol.

**Table 3 jcm-12-01934-t003:** The levels of indicators of cardiometabolic diseases at week 0, 12 and 52 of treatment with TNF-α inhibitors.

		Mean ± Standard Deviation or Median [Interquartile Range]			*p*		
Biologic (*n*)	Indicators	Week 0	Week 12	Week 52	Week 0 versus Week 12	Week 0 versus Week 52	Week 12 versus Week 52
IFX (28)	BMI (kg/m^2^) ^a^	25.54 ± 4.35	25.56 ± 4.36	25 ± 5.12	1.0	0.49	0.16
	HbA1c (%) ^a^	5.91 ± 0.52	6.06 ± 0.78	5.96 ± 0.85	0.65	1.0	1.0
	TC (mg/dL) ^a^	209.19 ± 41.27	212.29 ± 35.39	197.65 ± 38.44	0.52	1.0	0.89
	HDL-C (mg/dL) ^a^	40.42 ± 18.22	48.56 ± 20.22	41.78 ± 14.58	0.017 *	1.0	1.0
	LDL-C (mg/dL) ^a^	118.46 ± 34.62	146.67± 37.73	114.82 ± 38.49	1.0	0.45	0.63
	TG (mg/dL) ^b^	155 [103.5–213.75]	110 [79–170]	146 [86.5–271]	1.0	1.0	1.0
	UA (mg/dL) ^a^	6.289 ± 1.57	6.02 ± 1.5	5.48 ± 1.06	0.67	0.26	0.5
	sBP (mmHg) ^b^	128 [118–138.5]	128.5 [117.5–138]	128 [118–138.5]	1.0	0.6	0.41
	dBP (mmHg) ^a^	74.07 ± 11.21	72.92 ± 9.26	70.46 ± 10.84	1.0	1.0	1.0
ADA (15)	BMI (kg/m^2^) ^a^	23.8 ± 3.7	23.6 ± 3.72	23.16 ± 4.65	1.0	1.0	1.0
	HbA1c (%) ^a^	5.96 ± 0.73	5.83 ± 0.48	5.89 ± 0.55	0.86	0.61	1.0
	TC (mg/dL) ^a^	198.14 ± 39.14	215.92 ± 38.56	229 ± 45.51	0.48	1.0	1.0
	HDL-C (mg/dL) ^a^	58.64 ± 21.24	59.97 ± 22.75	64.9 ± 45.8	0.22	0.97	1.0
	LDL-C (mg/dL) ^a^	122.64 ± 45.54	128.7 ± 37.93	1.0	1.0	1.0	1.0
	TG (mg/dL) ^b^	93 [72–140.5]	89 [78–165]	219 [132.25–294.5]	0.19	1.0	1.0
	UA (mg/dL) ^a^	6.03 ± 1.04	5.39 ± 0.91	5.68 ± 0.65	0.048 *	0.157	1.0
	sBP (mmHg) ^a^	121.71 ± 21.4	120.82 ± 17.87	119.11 ± 22.13	1.0	1.0	1.0
	dBP (mmHg) ^a^	72 ± 14.82	70.73 ± 12.03	71.33 ± 12.56	1.0	1.0	1.0
CZP (11)	BMI (kg/m^2^) ^a^	25.17 ± 4.05	25 ± 4.4	25.11 ± 3.59	1.0	1.0	1.0
	HbA1c (%) ^a^	5.72 ± 0.3	5.82 ± 0.48	5.76 ± 0.25	0.65	1.0	1.0
	TC (mg/dL) ^a^	202.55 ± 33.36	209.36 ± 34.39	213.29 ± 40.1	0.52	1.0	0.89
	HDL-C (mg/dL) ^a^	65.5 ± 23.57	67.71 ± 21.55	68 ± 18.5	1.0	1.0	1.0
	LDL-C (mg/dL) ^a^	119.9 ± 26.56	124.22 ± 32.19	116.43 ± 31.12	1.0	0.45	0.63
	TG (mg/dL) ^b^	94 [80.5–139]	100 [92–113.5]	112 [82–140]	1.0	1.0	1.0
	UA (mg/dL) ^a^	6.05 ± 1.48	6.29 ± 1.29	6.21 ± 1.05	0.67	0.26	0.5
	sBP (mmHg) ^b^	120 [116–130]	125 [120–130]	120 [116–135]	1.0	0.6	0.41
	dBP (mmHg) ^a^	82.11 ± 14.1	80.78 ± 14.88	83.86 ± 19.5	1.0	1.0	1.0
All TNF inhibitors (56)	BMI (kg/m^2^) ^a^	24.96 ± 4.11	24.9 ± 4.21	24.6 ± 4.7	1.0	0.35	0.34
	HbA1c (%) ^b^	5.8 [5.5–6]	5.8 [5.5–6.15]	5.8 [5.48–6]	1.0	1.0	0.44
	TC (mg/dL) ^a^	204.73 ± 38.68	212.58 ± 35.25	207.32 ± 41.03	0.07	1.0	0.31
	HDL-C (mg/dL) ^a^	53.8 ± 23.09	58.65 ± 22.13	54.92 ± 26.06	0.009 **	0.868	1.0
	LDL-C (mg/dL) ^a^	120.24 ± 35.5	133.04 ± 36.1	122.61 ± 37.54	1.0	1.0	0.52
	TG (mg/dL) ^b^	120 [80.5–169]	100 [82.5–165]	145 [92–271]	1.0	1.0	0.84
	UA (mg/dL) ^b^	5.95 [5.43–6.9]	5.7 [5.03–11.2]	5.5 [5.05–6.4]	0.726	0.0046 **	0.537
	sBP (mmHg) ^a^	125.92 ± 17.4	126.87 ± 16.72	122.13 ± 18.05	1.0	0.23	0.48
	dBP (mmHg) ^a^	74.92 ± 12.99	73.94 ± 11.48	73 ± 13.64	0.68	0.61	1.0

^a^ Data provided as the mean ± standard deviation, analyzed by repeated measures analysis of variance. ^b^ Data provided as the median [interquartile range], analyzed by Friedman’s test. * Statistically significant at *p* < 0.05, ** at *p* < 0.01. BMI, body mass index; TC, total cholesterol; HDL-C, high-density lipoprotein-cholesterol; LDL-C, low-density lipoprotein-cholesterol; TG, triglyceride; UA, uric acid; sBP, systolic blood pressure; dBP, diastolic blood pressure.

**Table 4 jcm-12-01934-t004:** The levels of indicators of cardiometabolic diseases at week 0, 12, and 52 of treatment with IL-17 inhibitors.

		Mean ± Standard Deviation or Median [Interquartile Range]			*p*		
Biologic (*n*)	Indicators	Week 0	Week 12	Week 52	Week 0 versus Week 12	Week 0 versus Week 52	Week 12 versus Week 52
SEC (13)	BMI (kg/m^2^) ^b^	22.89 [21.26–29.41]	22.69 [20.71–24.8]	22.66 [20.66–26.81]	0.068	0.228	1.0
	HbA1c (%) ^a^	5.98 ± 0.5	6 ± 0.5	6.06 ± 0.64	1.0	1.0	1.0
	TC (mg/dL) ^a^	194.42 ± 31.60	203.55 ± 36.91	195.5 ± 24.69	1.0	1.0	1.0
	HDL-C (mg/dL) ^a^	71.4 ± 27.62	73.56 ± 31.6	56.35 ± 32.87	1.0	1.0	1.0
	LDL-C (mg/dL) ^a^	104.45 ± 25.08	109.89 ± 24.87	99.13 ± 18.67	1.0	1.0	0.21
	TG (mg/dL) ^b^	109 [81–131.5]	115 [90–128]	132.5 [87–197.5]	0.47	1.0	0.1
	UA (mg/dL) ^a^	6.11 ± 1.3	6.15 ± 1.55	6.35 ± 1.38	1.0	1.0	1.0
	sBP (mmHg) ^a^	126.33 ± 19.54	127.64 ± 15.67	121.6 ± 15.08	1.0	0.52	0.32
	dBP (mmHg) ^a^	71.42 ± 11.31	75.45 ± 9.9	78.7 ± 17.98	0.55	0.24	0.81
IXE (38)	BMI (kg/m^2^) ^a^	25.59 ± 4.72	24.78 ± 6.17	25.60 ± 4.25	1.0	0.36	0.058
	HbA1c (%) ^b^	5.7 [5.5–5.83]	5.65 [5.5–5.875]	5.7 [5.5–5.9]	1.0	0.33	0.29
	TC (mg/dL) ^a^	211.36 ± 37.55	207.07 ± 49.73	205.28 ± 39.72	1.0	0.57	1.0
	HDL-C (mg/dL) ^a^	51.87 ± 22.54	52.35 ± 29.83	46.03 ± 18.77	1.0	0.0089 **	0.4719
	LDL-C (mg/dL) ^a^	129.14 ± 31.35	125.38 ± 39.76	123.89 ± 37.73	1.0	0.92	1.0
	TG (mg/dL) ^b^	137.5 [91–200.75]	155 [97–197]	167 [119–218]	1.0	0.32	1.0
	UA (mg/dL) ^a^	5.83 ± 1.08	173.38 ± 125.49	5.91 ± 1.037	1.0	1.0	1.0
	sBP (mmHg) ^a^	125.31 ± 14	128.04 ± 18.64	125.07 ± 15.14	1.0	1.0	1.0
	dBP (mmHg) ^a^	80.16 ± 11.31	80.71 ± 11.47	79.73 ± 10.85	1.0	1.0	1.0
BRO (12)	BMI (kg/m^2^) ^a^	25.13 ± 3.22	24.94 ± 3.38	25.51 ± 3.11	1.0	1.0	1.0
	HbA1c (%) ^a^	5.81± 0.42	5.78 ± 0.41	5.73 ± 0.57	1.0	0.38	0.94
	TC (mg/dL) ^a^	188.5 ± 37.85	183.63 ± 21.37	192.14 ± 42.4	1.0	1.0	1.0
	HDL-C (mg/dL) ^a^	47.56 ± 20.82	53.03 ± 18.19	52.6 ± 23.09	0.34	0.56	1.0
	LDL-C (mg/dL) ^a^	120.11 ± 46.54	110.29 ± 20.96	119 ± 55.97	1.0	0.061	0.055
	TG (mg/dL) ^a^	121.1 ± 52.9	101 ± 41.41	126.71 ± 56.41	1.0	1.0	1.0
	UA (mg/dL) ^a^	7.03 ± 1.8	6.78 ± 1.66	6.5 ± 1.16	1.0	0.088	0.335
	sBP (mmHg) ^a^	124.11 ± 13.14	129.43 ± 12.8	124.13 ± 13.25	0.54	1.0	1.0
	dBP (mmHg) ^a^	74.22 ± 11.89	75.29 ± 10.35	75.5 ± 9.43	1.0	1.0	1.0
All IL-17 inhibitors (63)	BMI (kg/m^2^) ^a^	24.87 [22.36–29.87]	24.29 [21.57–28.39]	24.91 [22.38–28.54]	1.0	1.0	0.82
	HbA1c (%) ^b^	5.7 [5.5–5.9]	5.7 [5.5–5.9]	24.91 [22.38–28.54]	1.0	1.0	1.0
	TC (mg/dL) ^b^	200 [183–230.25]	194 [172–219]	199 [177.25–211]	0.48	0.53	1.0
	HDL-C (mg/dL) ^a^	54.77 ± 24.23	56.6 ± 29.44	49 ± 22.44	1.0	0.15	0.47
	LDL-C (mg/dL) ^b^	120.5 [98.25–139.75]	115 [93–132]	111 [90–134]	1.0	0.33	0.53
	TG (mg/dL) ^b^	131 [83.25–169.25]	119 [80.5–182]	141 [109.5–210]	1.0	0.66	0.5
	UA (mg/dL) ^a^	6.12 ± 1.35	6.16 ± 1.37	6.1 ± 1.13	1.0	0.98	1.0
	sBP (mmHg) ^a^	125.34 ± 14.99	128.15 ± 16.89	124.19 ± 14.59	1.0	1.0	0.26
	dBP (mmHg) ^a^	77.17 ± 11.82	78.63 ± 11.04	78.81 ± 12.25	0.42	0.66	1.0

^a^ Data provided as the mean ± standard deviation, analyzed by repeated measures analysis of variance. ^b^ Data provided as the median [interquartile range], analyzed by Friedman’s test. ** Statistically significant at *p* < 0.01. TC, total cholesterol; HDL-C, high-density lipoprotein-cholesterol; LDL-C, low-density lipoprotein-cholesterol; TG, triglyceride; UA, uric acid; sBP, systolic blood pressure; dBP, diastolic blood pressure.

**Table 5 jcm-12-01934-t005:** The levels of indicators of cardiometabolic diseases at week 0, 12, and 52 of treatment with IL-23 inhibitors.

		Mean ± Standard Deviation or Median [Interquartile Range]			*p*		
Biologic (*n*)	Indicators	Week 0	Week 12	Week 52	Week 0 versus Week 12	Week 0 versus Week 52	Week 12 versus Week 52
UST (10)	BMI (kg/m^2^) ^a^	25.08 ± 3.48	25.22 ± 3.47	24.93 ± 3.29	0.23	0.97	0.29
	HbA1c (%) ^a^	5.89 ± 0.32	5.73 ± 0.15	5.87 ± 0.32	0.62	1.0	0.5
	TC (mg/dL) ^a^	207.78 ± 37.76	226.86 ± 34.22	211 ± 46.38	0.13	1.0	0.53
	HDL-C (mg/dL) ^a^	43.7 ± 13.76	31.1 ± 16.12	27.8 ± 12.81	1.0	0.45	1.0
	LDL-C (mg/dL) ^a^	138.5 ± 39.63	140.5 ± 33.27	140.83 ± 37.40	0.052	0.552	1.0
	TG (mg/dL) ^a^	133.29 ± 39.77	184.67 ± 88.46	160 ± 75.98	0.57	1.0	0.66
	UA (mg/dL) ^a^	5.58 ± 1.26	6.27 ± 1.56	6.62 ± 1.63	0.88	0.66	1.0
	sBP (mmHg) ^a^	117.25 ± 14.38	113.86 ± 16.19	116 ± 20.4	1.0	1.0	1.0
	dBP (mmHg) ^a^	66.875 ± 9.28	71.14 ± 10.04	69 ± 13.14	0.2	1.0	1.0
GUS (19)	BMI (kg/m^2^) ^a^	25.33 ± 3.56	25.59 ± 3.6	25.66 ± 3.16	1.0	1.0	1.0
	HbA1c (%) ^a^	5.8 [5.6–6.4]	5.9 [5.55–7.08]	5.75 [5.6–6.03]	0.61	1.0	1.0
	TC (mg/dL) ^a^	199.58 ± 31.21	196.47 ± 34.53	192.94 ± 41.89	1.0	0.49	1.0
	HDL-C (mg/dL) ^a^	45.65 ± 20.66	49.09 ± 22.23	45.65 ± 14.71	1.0	1.0	0.87
	LDL-C (mg/dL) ^a^	122.7647 ± 31.84	116.38 ± 37.08	117.8 ± 39.93	0.89	1.0	1.0
	TG (mg/dL) ^b^	155 [108.5–196.5]	147 [102.5–197.5]	154.75 [93–154.75]	1.0	0.82	1.0
	UA (mg/dL) ^a^	5.74 ± 1.5	5.87 ± 1.36	135.38 ± 52.59	1.0	1.0	1.0
	sBP (mmHg) ^a^	127.39 ± 14.9	126.39 ± 15.64	128.63 ± 14.48	1.0	1.0	1.0
	dBP (mmHg) ^a^	78.89 ± 12.61	77.78 ± 10.13	78 ± 13.06	1.0	1.0	1.0
RIS (17)	BMI (kg/m^2^) ^a^	24.27 ± 3.84	24.02 ± 3.95	24.48 ± 4.29	1.0	0.82	0.67
	HbA1c (%) ^a^	5.85 ± 0.76	5.75 ± 0.41	5.74 ± 0.45	1.0	0.65	0.88
	TC (mg/dL) ^a^	202 ± 25.42	196.85 ± 30.30	183.55 ± 24.40	0.13	1.0	0.53
	HDL-C (mg/dL) ^a^	56.09 ± 31.43	43.6 ± 25.15	43.36 ± 20.92	0.78	1.0	1.0
	LDL-C (mg/dL) ^a^	118.93 ± 36.15	120.38 ± 37.79	113.27 ± 28.17	0.052	0.552	1.0
	TG (mg/dL) ^a^	146.56 ± 77.33	182.46 ± 143.67	134.55 ± 77.59	0.57	1.0	0.66
	UA (mg/dL) ^a^	6.38 ± 1.68	6.24 ± 1.34	6.06 ± 1.19	0.88	0.66	1.0
	sBP (mmHg) ^a^	136.4 ± 23.41	133 ± 18.71	134.54 ± 13.97	1.0	1.0	1.0
	dBP (mmHg) ^a^	84.07 ± 17.76	82.27 ± 14.11	82.62 ± 10.28	0.2	1.0	1.0
All IL-23 inhibitors (46)	BMI (kg/m^2^) ^a^	24.84 ± 3.65	24.96 ± 3.63	25.05 ± 3.59	1.0	1.0	1.0
	HbA1c (%) ^b^	5.8 [5.6–6]	5.8 [5.5–6.1]	5.7 [5.6–6]	0.64	0.66	1.0
	TC (mg/dL) ^a^	202.14 ± 30.13	202.05 ± 34.32	194.58 ± 39.02	1.0	0.12	0.23
	HDL-C (mg/dL) ^a^	49.46 ± 24.77	44.49 ± 22.88	41.52 ± 17.64	1.0	0.84	1.0
	LDL-C (mg/dL) ^a^	122.97 ± 34.37	120.88 ± 36.63	120.56 ± 36.10	1.0	1.0	1.0
	TG (mg/dL) ^b^	142.5 [98.5–193]	149 [105.25–199.75]	127 [95.5–157.5]	0.35	0.37	0.19
	UA (mg/dL) ^a^	5.94 ± 1.53	6.08 ± 1.36	6.13 ± 1.52	1.0	0.85	0.96
	sBP (mmHg) ^a^	128.71 ± 19.28	126.16 ± 17.64	127.97 ± 16.76	0.89	1.0	1.0
	dBP (mmHg) ^a^	78.44 ± 15.22	77.86 ± 11.8	77.68 ± 12.86	0.73	1.0	1.0

^a^ Data provided as the mean ± standard deviation, analyzed by repeated measures analysis of variance. ^b^ Data provided as the median [interquartile range], analyzed by Friedman’s test. TC, total cholesterol; HDL-C, high-density lipoprotein-cholesterol; LDL-C, low-density lipoprotein-cholesterol; TG, triglyceride; UA, uric acid; sBP, systolic blood pressure; dBP, diastolic blood pressure.

**Table 6 jcm-12-01934-t006:** Correlations between percent reduction of PASI versus percent changes of HDL-C or UA.

		Week 12		Week 52	
Biologic (*n*)	Indicators	Rho	*p*	Rho	*p*
Infliximab (28)	HDL-C	0	1.0	NA	
Adalimumab (15)	UA	0.309	0.305	NA	
Ixekizumab (38)	HDL-C	NA		0.105	0.633
All TNF-α inhibitor (56)	HDL-C	−0.205	0.372	NA	
All TNF-α inhibitor (56)	UA	NA		0.146	0.467

Correlations between variables were examined using Spearman’s correlation coefficients. HDL-C, HDL-C, high-density lipoprotein-cholesterol; UA, uric acid; TNF-α, tumor necrosis factor-α; NA, not applicable.

## Data Availability

Not applicable.

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
