# Peer review of "Effects of Biologic Therapy on Laboratory Indicators of Cardiometabolic Diseases in Patients with Psoriasis"

_jcm, 2023, doi:10.3390/jcm12051934_

Round 1

Reviewer 1 Report

Hagino and co-workers intended to examine the effects of various biologics on serum chemistry tests related to cardiometabolic diseases in patients with psoriasis. The aim of the study is important for the understanding of comorbidities of psoriasis. However, the manuscript needs major revision. Please answer to comments below and revise the manuscript accordingly.

[Major Comments]

1. I would strongly advise “NOT” to write speculation in the abstract that is not adequately supported by the discussion in the text. The final sentence in the abstract is an overstatement and should be deleted. One cannot say that TNF-alpha inhibitors may reduce the risk of CVD through an increase of HDL-C levels and a decrease of UA levels when the results presented are not consistent within different TNF-alpha inhibitors, especially when sample size is different between biologics. Furthermore, the authors showed that HDL-C levels of patients treated with IXE were lower than baseline at week 52, but it was ignored in the last line. Additionally, the authors were only able to show the results in week 12 or week 52, and the results of two time points are also inconsistent. I believe that authors should revise the abstract and restrict themselves to describing the results of their study instead of writing unsupported speculation.

2. There seem to be considerable differences between what authors found in serum chemistry tests for IFX and ADA, but the discussion is sparse. Authors should discuss any reasons for discrepancies between serum chemistry tests of IFX and ADA, although they are both TNF-alpha inhibitors. Additionally, is there any discussion on CZP? 

3. Can a transient decrease of HDL-C or UA levels at week 12 that is back to baseline in week 52 be protective against cardiometabolic disease? Are there any published results from long-term follow-ups of clinical trials suggesting changes in those serum chemistry data after more than one year to support the authors’ expectations?

4. Did the authors investigate new development of DM, HTN, DL, and HUA during the study period? If they did, was there any difference between the three classes of biologics?

5. The inclusion and exclusion criteria must be stated clearly in the methods section. Did all psoriasis patients treated with biologics in this hospital continue at least 1 year of treatment with the same agent? I am suspicious because there must be some patients who stop treatment or change to different agents. If they excluded any patients, the authors should mention it.

6. Were all patients naïve to biologics? If some patients had history of biologics treatments, it could affect the laboratory data and become a confounder. Please discuss this point in the methods and discussion or limitations.

7. Table 1 needs substantial improvement. The table is tough to read.

- Do not change the notation within each row. Some rows contain both cells shown in means and cells shown in median. For example, the row for age has superscript “a”, but the cell for GUS is shown in the median. 

- Either male or female row is sufficient to show the gender ratio.

- It might be better to split table 1 into two tables. For example, one table can compare TNF-alpha inhibitors combined, IL-17 inhibitors combined, and IL-23 inhibitors combined while the other table can list individual biologics. In this way, it is easier to understand the difference in demographics and characteristics between each class of biologics.

- The result section for Table 1 is poor. It only tells the number of patients included in the study. Consider revising the section.

8. Authors need to consider significant digits/figures when they present their results. With only 165 samples, I do not expect three significant figures for each cell. For example, age should be shown as median with an integer, and percentage should be shown as an integer or one decimal place. Include a 0 in the cells (e.g., 1.0 instead of 1) where appropriate.

9. I need help understanding what figure 2 is meant to convey to readers. First, if the authors want to show the median of any value in a figure, they should use a box plot or add IQR values to the bar graph. Second, there seems to be some difference in terms of median percent reduction of PASI, at least between IL-17 inhibitors and UST, but the authors failed to point this out. Actually, the difference between PASI improvement and each biologic is illustrated in figure 1 and the authors mentioned it in the text. Therefore, figure 2 and the accompanying text needs to be more accurate.

In summary, figures 1 and 2 are redundant. Figure 1 contains information that readers need, while figure 2 is misleading. I would advise to simply delete figure 2 for simplicity.

[Minor comments]

10. Are there no conflicts of interest, for example honoraria or research funding from a pharmaceutical company selling biologics studied in this research, for all authors? If there is, those COI should be disclosed. Please disregard this comment if there is actually no COI to disclose.

11. The term “Whole TNF-alpha inhibitors” that the authors used is strange. I have never seen this term used in literature, and the sentences which use “whole TNF-alpha inhibitors” are difficult to follow. Rather than using “whole”, the authors should better use other words that match the context, for example 

- “all” TNF-alpha inhibitors

- “any one of the” TNF-alpha inhibitors 

- TNF-alpha inhibitors “combined”

12. I do not find clinical indicators of cardiometabolic diseases discussed in this manuscript. BMI and blood pressure are not mentioned in the result section or the discussion section at all. Listing these data in the table is no use if the authors do not discuss it. Thus, the title of the manuscript can delete “clinical indicator” and make it simple as “Effects of biologic therapy on laboratory indicators of cardiometabolic diseases in patients with psoriasis.”

13. From the way the authors present Table 1 and other tables, I need clarification about the statistical analysis they performed. Please review the analysis again to confirm that the authors have conducted the analysis correctly. You may consider inviting epidemiologist or biostatistics specialist as a co-author. 

Author Response

添付資料をご覧ください。

Reviewer 2 Report

The original article titled: The effects of biologic therapy on laboratory and clinical indicators of cardiometabolic diseases in patients with psoriasis consists of 5 typical parts: an abstract, an introduction, material and method, discussion and references.

Both the abstract and the introduction is concise. They clearly indicate the problem of the paper - the impact of therapies used in psoriatic diseases on cardiometabolic diseases and its nature - a retrospective study.

Eleven therapies used in Japan were selected for the analysis and a total of 165 patients were analyzed with a clear difference in terms of gender - 80% were male. What could have caused such a difference? Was it due to the selection of patients into particular groups? Are men more likely to get psoriasis in Japan? There are no such differences in Europe. Maybe it's worth mentioning.

The results consist of 7 subsections, each of which presents a different problem. There is a descriptive part, each time supported by a very detailed table. I only have a problem with the readability of the PASI75/90/100 charts.

The discussion is correct. I appreciate that it was enriched by the authors with the weaknesses of the study.

References include 35 items, including 7 from the last 3 years. In pubmed I found two more articles from 2020 that can be analyzed and possibly added to References: doi: 10.3390/medicina56090473 and doi: 10.1111/dth.14153.

Round 2

Reviewer 1 Report

The authors have responded to most of my comments in the previous round of review. Please consider several comments listed below before reaching final decision.

1) Once again, I need to ask the authors whether there was no patient at all in the authors’ clinics who discontinued the biologics treatment within 1 year. I assume that there must be some patients who were not able to continue a treatment with single biologics for one year. In this case, lines 52–54 is incorrectly describing the inclusion and exclusion of the patients. The manuscript tells us that 165 is the total number of patients that was treated in the authors’ clinics during the study period.  

I would be surprised if the authors’ clinics always continue one biologics treatment for at least 1 year for all for patients under any conditions or adverse events. If the authors only selected patients who were treated with one of the biologics for at least 52 weeks, and excluded patients who stopped those treatments before 52 weeks, write down the criteria explicitly for the readers.

2) Authors responded to most of my comments on tables, but they have misunderstood some of my suggestions in the previous round of review.

-       Authors must include IQR for age. The median can be as it is in the revised manuscript.

-       Disease duration and PASI are shown in median (IQR) in Table 1a. Those valuables for most of biologics are shown in median (IQR) in Table 1b as well. I would advise the authors use median (IQR) for all biologics in Table 2 because we expect nonparametric distribution for those variables and for comparison between biologics.
